# Polymer Coating Integrity, Thrombogenicity and Computational Fluid Dynamics Analysis of Provisional Stenting Technique in the Left Main Bifurcation Setting: Insights from an In-Vitro Model

**DOI:** 10.3390/polym14091715

**Published:** 2022-04-22

**Authors:** Marek Milewski, Chen Koon Jaryl Ng, Pawel Gąsior, Shaoliang Shawn Lian, Su Xiao Qian, Shengjie Lu, Nicolas Foin, Elvin Kedhi, Wojciech Wojakowski, Hui Ying Ang

**Affiliations:** 1Division of Cardiology and Structural Heart Diseases, Medical University of Silesia in Katowice, 40-635 Katowice, Poland; marek.milewski92@gmail.com (M.M.); p.m.gasior@gmail.com (P.G.); ekedhi@yahoo.com (E.K.); wojtek.wojakowski@gmail.com (W.W.); 2National Heart Research Institute Singapore, National Heart Centre Singapore, 5 Hospital Drive, Singapore 169609, Singapore; jaryl.ng.chen.koon@gmail.com (C.K.J.N.); lu.shengjie@nhcs.com.sg (S.L.); nicolas.foin@gmail.com (N.F.); 3Department of Biomedical Engineering, National University of Singapore, Singapore 119077, Singapore; lianshaoliang@u.nus.edu; 4Division of Chemical and Biomolecular Engineering, Nanyang Technological University, Singapore 637459, Singapore; suxiaoqian@gmail.com; 5Duke-NUS Medical School, Singapore 169857, Singapore; 6Erasmus Hospital, Université libre de Bruxelles (ULB), 1070 Brussels, Belgium

**Keywords:** polymers damage, drug-eluting stent, side branch ostia thrombogenicity, bifurcation lesions

## Abstract

Currently, the provisional stenting technique is the gold standard in revascularization of lesions located in the left main (LM) bifurcation. The benefit of the routine kissing balloon technique (KBI) in bifurcation lesions is still debated, particularly following the single stent treatment. We compared the latest-generation drug-eluting stent (DES) with no side branch (SB) dilatation “keep it open” technique (KIO) vs. KBI technique vs. bifurcation dedicated drug-eluting stent (BD-DES) implantation. In vitro testing was performed under a static condition in bifurcation silicone vessel models. All the devices were implanted in accordance with the manufacturers’ recommendations. As a result, computational fluid dynamics (CFD) analysis demonstrated a statistically higher area of high shear rate in the KIO group when compared to KBI. Likewise, the maximal shear rate was higher in number in the KIO group. Floating strut count based on the OCT imaging was significantly higher in KIO than in KBI and BD-DES. Furthermore, according to OTC analysis, the thrombus area was numerically higher in both KIO and KBI than in the BD-DES. Scanning electron microscopy (SEM) analysis shows the highest degree of strut coating damage in the KBI group. This model demonstrated significant differences in CFD analysis at SB ostia with and without KBI optimization in the LM setting. The adoption of KBI was related to a meaningful reduction of flow disturbances in conventional DES and achieved results similar to BD-DES.

## 1. Introduction

Overall, 15–20% of all percutaneous coronary interventions (PCI) take place in bifurcation lesions [1]. They predispose to recurrent, unfavorable complications like in-stent restenosis (ISR) and, in particular, stent thrombosis (ST), which could lead to increased mortality. The clinical data shows that about 23% of all ST occurs in lesions located in bifurcations [2]. Therefore, the optimization of the bifurcation PCI technique is a subject of bench studies and clinical trials.

The challenge of performing interventions in bifurcation lesions is associated with their morphological complexity. Furthermore, the diameter differences between the distal and proximal vessel and side branch (SB) require the proper implantation technique to reach the suitable apposition of the drug-eluting stent (DES). It consists of proximal optimization (POT) and SB optimization. Moreover, lesions located in bifurcation are susceptible to lagging the arterial healing after PCI. Better vascular healing occurs in the newest generation of DES, so therefore they should be considered as the preferred stent choice [3].

In first-generation durable-polymer DES (DP-DES), we can observe a lower number of restenosis with a higher amount of ST than in bare-metal stents (BMS) [4]. The introduction of second-generation DP-DES resulted in lower amounts of restenosis and ST [5]. However, very late ST and neoatherosclerosis were still observed; this is probably due to vessel reaction to the presence of durable polymer [6]. To reduce this potential adverse outcome, a biodegradable polymer coated platform (BP-DES) was developed. Due to the hydrolysis of the ester bonds, their coating degrades into lactic or glycolic acids, which–as a result of tricarboxylic acid cycle–slowly degrades into carbon dioxides and water, secreted by the respiratory system [7]. BP-DES were developed to obtain the antiproliferative properties of DP-DES and finally become a BMS. However, the large meta-analysis showed that there were no significant differences in long-term outcomes in DP-DES and BP-DES [8]. Both platforms have similar safety and efficacy [9].

The latest data from large trials demonstrated that in low- and intermediate-risk patients, the results of stenting the left main coronary artery (LMCA) are comparable to coronary artery bypass grafting [10,11,12]. The huge part of the left ventricular myocardium is supplied by the LMCA, so the atherosclerotic stenosis in LMCA is related to meaningful myocardial danger. Furthermore, the suboptimal stenting technique used in the LMCA interventions translates into a high incidence of adverse events [13]. One of the open questions is whether the routine stent strut opening with kissing-balloon inflation (KBI) in SB is necessary for the absence of SB stenosis. Recently published data from the randomized EBC MAIN study demonstrated similar outcomes in patients treated with a provisional stenting technique implementing a single stent usage, compared to patients treated with a more complicated two-stent strategy [14]. Therefore, it seems that provisional stenting should remain as the gold standard in most LMCA true bifurcation procedures. Nevertheless, the ideal treatment strategy for SB protection remains unclear and challenging. The potential benefits of KBI compared with no the SB dilatation “keep it open” method (KIO) in bifurcation lesions is still debated, particularly following the single stent treatment [15,16].

It is also necessary to make out the hemodynamic parameters inside bifurcation because every disruption of physiological flow could increase the inflammation degree and promote plaque progression. The prothrombotic environment occurs as a result of protein adhesion, and blood platelets activation and aggregation close to the vessel wall [17,18,19]. Furthermore, the struts that overhang at the ostium of the SB are probably a relevant point for thrombus formation, and therefore could lead to ST [20]. Benchtop flow loop systems set by various groups showed the advantages and reproducibility of this method to rate different stents and stenting methods. It provides ex vivo data on the impact of different stents design, strut thickness, and implantation techniques on acute thrombogenicity evaluation. Benchtop tests also allow to evaluate the impact of flow disruption following device implantation, and potentially could facilitate development of new stent platforms. In addition, it could theoretically improve the selection of specific implantation techniques in individual lesion subsets in order to reduce the rate of complication following angioplasty [21,22,23,24,25]. 

In this investigation, we wanted to liken the thrombogenicity, flow disturbances, and polymer coating damages at the ostium of SB after new-generation DES implantation with KBI and KIO approach, versus bifurcation dedicated stent (BD-DES) with the usage of an in-vitro bifurcation model. 

## 2. Materials and Methods

### 2.1. Device Description

In this experiment, we used the latest-generation conventional DES (Xience Sierra, Abbott, Santa Clara, CA, USA) and BD-DES (Bioss LIM C, Balton, Warsaw, Poland). Each tested platform is accessible commercially in countries with CE marks. The newest generation Xience Sierra is an everolimus-eluting stent from the widely used Xience family. Xience Sierra was designed as a cobalt-chromium (Co-Cr) platform, with the strut thickness equal to 81 µm. Everolimus is mixed with undegradable acrylic and fluoropolymers. Xience Sierra belongs to a durable polymer DES group, in which the polymer coating stays permanently on the stent surface after the release of the drug [26]. The polymer coating consists of acrylic and fluoropolymers (vinylidene-fluoridehexafluoropropylene copolymer). Furthermore, the Co-Cr platform is used in the BD-DES, Bioss LIM C, and it has a strut thickness equal to 70 μm. It is a sirolimus-eluting stent, where the drug is eluted from a biodegradable coating made up of glycolic acids copolymer and lactic. Degradation of the polymer lasts eight weeks [27]. The device is delivered on a quick-replacement catheter with a semi-compliant (SC) balloon. The BD-DES is made of two major parts, both with various diameters. To provide the physiological compatibility and correct flow conditions by minimizing the flow disturbances–as all struts are well-apposed (WA) to the wall of the vessel–the ratio between the diameter of proximal and distal part is between 1.15 to 1.3. The platform consists of a central region with two connecting struts, sized 2.0–2.4 mm. For each platform, the polymer coating is applied by manufacturers during the production phase, with the usage of their proprietary technologies. Specifically for this experiment, all platforms were bought from particular manufacturers.

### 2.2. Deployment of the Stents

In this study, we compared BD-DES, Bioss Lim C (4.25–3.50 × 25 mm, *n* = 5), and Xience Sierra (3.5 × 24 mm) implanted with two techniques: KIO (*n* = 5) and KBI (*n* = 5). We implanted all stents into a Y-shaped silicone model of a bifurcation (diameters: proximal segment = 5.5 mm, distal segment = 3.5 mm and SB = 3.5 mm, fabrics: Shore 40A Silicone, the angle between main branch (MB) and SB equals 90°) (Figure 1).

According to clinical measurements of the angle between the left anterior descending artery (LAD) and circumflex artery (Cx), the angle between the MB and SB in the model was similar to clinical conditions [28,29]. The angles between branches and the proximal part of the model were set at 135°. Stents were firstly implanted at the pressure specified by the manufacturers to reach intentional stent diameter. Then, to provide proper apposition of the struts in the proximal region of the bifurcation model, the POT was performed with a 5.5 mm SC balloon inflated to 14 atm (5.78 mm according to the compliance chart). Subsequently, in the KBI group, two 3.5 mm non-compliant (NC) balloons were inflated to nominal pressure in both the distal MB and SB followed by a final POT using a 5.5 mm SC balloon inflated to 14 atm. In all groups, OCT pullback (C7x OCT Imaging System, LightLab Imaging Inc., Westford, MA, USA) was obtained just before blood perfusion. The speed of pullback was 10.0 mm/s, with a length of pullback equal 54.0 mm (the equivalent of 540 frames) and a resolution of 15 µm. In Figure 2, there is a flow chart that describes the study plan.

### 2.3. Flow Perfusion

To perfuse fresh porcine blood with the addition of 10% anticoagulant (consists of acid-citrate-dextrose) from the reservoir to implanted stent and back, a special peristaltic pump (Minipuls3, Gibson, USA) was used. The pump provided a continuous flow rate of 200 mL/min for one hour. The perfusion was performed following previous studies, and was similar to the coronary arteries’ blood flow [21]. During perfusion, porcine blood was heated to the physiological temperature of 37 °C. To eliminate remaining blood before performing OCT imaging, all models were rinsed with Tyrode’s dilution after 60 min. Following OCT examination, all stents were gently pulled out from the bifurcation model and examined with SEM imaging analysis. We took extra care during removal to minimalize any contact with in-stent clots and to avoid any loss.

### 2.4. OCT Analysis

Before the blood perfusion, we performed the OCT imaging of the proximal region to measure important dimensions. The ratio between the maximal and minimal diameter of each analyzed OCT frame (D_max_/D_min_ at 1 mm intervals, every ten frames with a pullback speed of 10 mm/s), standardized for the length of the stent, gives the elliptical index of the proximal part [30]. To estimate the area of thrombus in the bifurcation region, the OCT cross-section analysis was performed for each frame. All struts in the opening angle of the SB were described as floating struts. The struts that protruded into the lumen of the vessel at a distance greater than the thickness of the strut were described as malapposed (MA) [31]. For the region of bifurcation ostium, the quantity and percentage of WA, MA, and floating struts were calculated. To compute the area of the thrombus for each sample, we selected and averaged three OCT frames with the most extensive thrombus area in the bifurcation region. A qualified operator manually performed all OCT measurements.

### 2.5. Computational Fluid Dynamics

To parse the flow patterns and shear rate, the CFD was performed based on a previously established method [20]. The methodology uses 2D longitudinal images from OCT pullback with boundary conditions based on experimental flow to quantify shear rate within the bifurcation model. It allowed identifying segments of the struts with a greater flow disturbance risk caused by floating struts and MA. Based on OCT images, the 2D longitudinal geometries of stented models were recreated. Subsequently, they were meshed and then, with the usage of fluid computational software (Fluent, ANSYS), simulated with flow conditions similar to experimental conditions. A quadrilateral dominant mesh was assigned to the models with a maximum element size of 0.1 mm, with finer mesh along the boundaries of the vessel wall as well as stent struts to capture more detail on the shear rates near vessel walls and stent strut surfaces. A mesh convergence study was done to ensure that further refinement of element size would not greatly change the centre-line flow velocity of the model, and hence accuracy of the shear rate prediction. There were about 95,000 elements per model. Boundaries of the model were created as inflexible with the no-slip condition. Blood was modeled as Newtonian, incompressible liquid. The density of blood was equal to 1060 kg/m^3^ and stickiness of 0.0035 Pa.S [32]. At the entry of the bifurcation model, a boundary condition of the flow speed of 0.14 m/s was implemented; this was similar to the experimental flow rate equal to 200 mL/min. The outlets in the model were assigned as zero pressure [33]. We obtained the high shear rate area (>1000 s^−1^) physiologically; in regular human arteries the flow rate falls within the scope of 100–1000 s^−1^) [34]. Additionally, we obtained the maximum shear rate from this experiment.

### 2.6. Drug Coating Integrity

All stents, after performing the blood flow perfusion, were softly pulled out from the bifurcation model with the usage of two guide wires. Before that, the models were pressed tenderly at the proximal and distal parts so as to simplify the process and avoid causing any distortion to the main part of the stent. To pull out the stent as delicately as possible, a tweezer was used at one end of the model. The same protocol of removal was used for all stents. In this part of the study, we did not include the proximal and distal ends of the stent. With the usage of SEM, all stents were then examined. All events of strut coating damage were counted and then qualified into four groups in accordance with damage grade. Qualification into each group was performed in accordance with a formerly published study [35].

### 2.7. Statistical Analysis

SigmaStat software (version 4.0; Systat Software, San Jose, CA, USA) was used to make the statistical analysis. Results are shown as median (interquartile range). Values were assumed to be nonparametric. All values were analyzed with Kruskal-Wallis one-way ANOVA because of a low count of samples. The Dunn’s test that compared KIO, KBI, and BD-DES groups was performed post hoc when the ANOVA was significant. The differences were only considered meaningful if the received *p*-value was <0.05,.

## 3. Results

### 3.1. Optical Coherence Tomography

In Table 1 we presented a recapitulation of the OCT analysis. The percent of WA struts was meaningly higher in the KBI and BD-DES than in KIO (respectively: 93.29 (88.06–93.39)% vs. 94.29 (88.46–98.81)% vs. 78.64 (76.07–85.28)%, *p* = 0.28). The percentage of floating struts based on the OCT imaging was significantly higher in KIO than in KBI and BD-DES (KIO: 15.93 (12.99–17.65) vs. KBI: 3.84 (3.46–5.06) vs. BD-DES: 0 (0–2.86), *p* = 0.33). Additionally, the percentage of MA struts was significantly lower in the BD-DES. Thrombus area was numerically lower in BD-DES when compared to the KBI and KIO group (KIO: 0.52 (0.17–0.65), KBI: 0.70 (0.15–1.16), BD-DES: 0 (0–0.09), *p* = 0.15). Representative OCT images, percentage of floating strut, and thrombus quantification are shown in Figure 3.

### 3.2. Computational Fluid Dynamics

The high shear rate (>1000 s^−1^) area was, by the OTC images, statistically higher in the KIO group when compared to KBI (respectively: 0.12 mm^2^ (0.09–0.12) vs. 0.02 mm^2^ (0.01–0.023) *p* = 0.0133). Furthermore, maximal shear rate was higher in number in the KIO group (KIO: 2293 s^−1^ (2072–2365) vs. KBI: 1621 s^−1^ (1299–2230) vs. BD-DES: 1375 s^−1^ (1348–2068), *p* = 0.14). Images from CFD analysis and the areas of high and maximum shear rate are shown in Figure 4.

### 3.3. Drug Coating Integrity

A recapitulation of the SEM analysis of drug coating integrity is presented in Table 2. We observed the most significant strut coating damage in the KBI group. BD-DES group displayed the lowest number of second category coating damages when compared to KBI (1 (0–3) vs. 21(18–21) *p* = 0.0026), third category when compared to KIO and KBI (0 (0–1) vs. 4 (3–12) *p* = 0.046 vs. 11 (7–12) *p* = 0.0103), and fourth category when compared to KBI (0 vs. 20 (16–21), *p* = 0.0024). Representative SEM images of drug coating damage are presented in Figure 5.

## 4. Discussion

In this study, with the usage of the in-vitro bifurcation model, we compared the thrombogenicity, flow disturbances, and polymer coating damages at the ostium of SB after new-generation DES implantation, using KBI and KIO techniques versus BD-DES. The main outcomes of this study are that (1) in KBI and BD-DES groups there was a significantly higher percentage of well-apposed struts than KIO, (2) BD-DES had a significantly lowest percentage of MA struts, (3) in BD-DES there was a numerically lower thrombus area, (4) KBI group have statistically more coating damage, and (5) KIO is associated with the statistically higher area of high shear rate.

Interventions in bifurcation lesions are difficult to perform due to their morphological complexity and are related to a higher number of unfavorable clinical events like ST and ISR [36]. The difference between the diameter of the proximal and distal part of the vessel requires stent sizing according to the distal diameter, and optimalization of the proximal part with the usage of the larger balloon to achieve proper expansion. Furthermore, delayed arterial healing occurs in bifurcation lesions [37].

According to recently published data, provisional stenting is recommended as a golden standard in treating LM bifurcations since it brings favorable outcomes when compared to the two-stent technique. A randomized EBC MAIN study shows that in LM bifurcation stenosis–which requires angioplasty–a lower number of major adverse cardiac events occurred in provisional stenting than in dual stent strategy [14]. However, it is vague when it comes to outlining the ideal strategy for protecting SB and the benefits of performing KBI compared with KIO are still debated.

In vitro bench testing is necessary to comprehend the hemodynamic parameters inside the bifurcation, which is important to evaluate the potential benefits of the KBI and KIO method compared to BD-DES. The overhanging struts in SB ostium could be the main spot for thrombus formation and could lead to ST [20]. A previously published study with the same bifurcation model showed a correlation between struts protruding into SB ostium and increased thrombogenicity [23]. The study shows that the floating struts have an influence on thrombus formation.

Likewise, according to previous publications, most ST events in patients who underwent PCI procedure were associated with morphological abnormalities in OCT imaging, such as strut malposition and underexpansion. Furthermore, MA was the main cause of late and very late ST [38,39]. In our study, the KIO group had a significantly lower percentage of WA struts and a significantly higher percentage of floating struts than KBI and BD-DES, with a meaningly lower percentage of MA struts in BD-DES. This may explain that the area of thrombus was numerically lower in the BD-DES than in the KIO and KBI.

Plaque formation and neointimal hyperplasia are promoted by any disturbance of the laminar flow in the vessel [40]. Abnormal flow patterns could lead to ST [41], even with inhibition of neointimal growth by the antiproliferative drug elution from the stent surface. Previous studies showed that high share rate regions related to plaque erosion and ruptures could lead to aggregation of platelets and near the wall of the artery [42].

In the LMCA bifurcation, the proximal part of the stent is often overexpanded [43]. The forces that act on the stent struts could lead to polymer coating damage, and as a result deformation of the platform; probably a smaller thrombus area in the BD-DES relates to the stent architecture that composes of just two connecting struts at the ostium of SB. After balloon inflation, the “self-positioning” of this part occurs so that the side branch can “stay open” [44]. In the BD-DES, two connecting struts reduce a metal-to-artery ratio, but accordingly to a previously published study, it does not impact the outcomes in the LMCA PCI [45].

The presence of struts at SB in bifurcation lesions might disrupt blood flow and increase the high shear rate regions. At least three different mechanisms of platelet aggregation connected to shear rate have been identified [46]: a low shear rate that normally occurs in veins and arteries, a high shear rate that can be found in microcirculation and in moderately narrowed arteries, and the maximal shear rate in severe arterial stenosis. The occurrence of high shear rate areas is related with the matrix metalloproteinase activation, which leads to phenotypic conversion into features of vulnerability of the plaque [47,48].

As a result, it is linked with regression of fibrous and fibrofatty tissues and progression of expansive remodeling, calcium, and the plaque necrotic core. Additionally, in the high shear rate areas in the proximal parts of lesions, the features of a high-risk plaque tended to be localized [49]. According to these observations, recent studies show that the high shear rate area in the proximal part of lesions were related with a higher risk of acute coronary syndromes [50,51]. In this examination, CFD analysis showed a statistically higher area of high shear rate in KIO than in KBI and BD-DES. Furthermore, the maximum shear rate was the highest in number in KIO; this might explain the numerically higher thrombus area in KIO based on IF and OCT analysis. Previous studies using CFD have also drawn associations between larger flow disturbance with increased acute thrombogenicity [19,32], thus corroborating with the results derived from the CFD models and experimental setup in this study. Recently, the European Bifurcation Club consensus recognized the value of in vitro bench testing and computational simulations in bifurcation interventions that help improve knowledge within this field, indicating potential value of this platform to answer other clinically relevant questions using benchtop setup and CFD modelling [52].

Moreover, the drug coating damage or detachment could increase the inflammation with neointimal reaction and lead thrombogenic factor to ST [36,53]. Pieces of polymer peeled from DES surface may cause coronary microembolism and inflammation. In our SEM analysis, we observed the smallest degree of coating damages in BD-DES and the highest in KBI. This could be attributed to the unique BIOSS LIM C architecture, with the proximal part of the stent larger (by 0.75 mm) than in other, conventional DES. This entails the lower overexpansion of the stent and consequently reduces strut coating damage. Furthermore, KBI technique is associated with increased polymer coating damages, especially in the ostial regions of devices due to greater overexpansion and forces exerted on the struts when compared to KIO. This might potentially impact drug elution process in the regions with damaged polymer, and subsequently lead to uneven drug distribution in the arterial wall, which could theoretically increase the risk of ISR [43]. Therefore, in vitro studies are crucial in order to improve the polymer coating technology, which could translate into reduction of device-related adverse events rates

## 5. Conclusions

This study shows that bench testing of bifurcation in vitro model is a useful instrument and could help improve bifurcation angioplasty. This model showed differences in thrombogenicity and mechanical properties at SB ostium when two different ways of protecting SB were used. It suggested the advantages of the BD-DES and KBI method compared with KIO. The adoption of KBI was related to a meaningful reduction of flow disturbances in conventional DES, and achieved results comparable to the BD-DES. This study supports previous publications using this model, which provided an extensive in vitro evaluation of the different stent types and implantation techniques performed in the bifurcation model. In the future, this model can be useful to evaluate different devices, implantation techniques, and various bifurcation geometries [54].

## 6. Limitations

This study has some limitations typical of static in vitro benchtop experiments. The results of this study were obtained in a non-stenotic vascular model, which does not reflect the atherosclerotic lesion in a clinical setting. Furthermore, the simplified left main model does not completely show the human tissues’ response to stenting and other complexities of bifurcation anatomies. Hence, the data should be interpreted carefully and should be understood as an approximation of the real behavior of the stenting artery response while overstretching [55]. In our model, the perfusion was constant instead of a pulsating flow that occurs physiologically, which could have had an influence on the study results. Nevertheless, available literature shows that usage of constant flow was acceptable as thrombus formation is more affected by the shear generated than flow pulsatility [56].

This benchtop model only shows acute thrombogenicity of stent deployment [21,23], and in the future animal testing should be conducted to understand completely the long-term stent thrombogenicity. Although 2D models are incapable of completely capturing differences between models with various stent construction, it has been shown previously that they demonstrated shear rate trends consistent with experimentally thrombogenicity outcomes [19,57]. Future improvements can include creating 3D CFD models; this could help to reconstruct patient-specific coronary anatomy [58] and complex strut configurations–even for models with more than one stent [59]–for more precise hemodynamic results.

## Figures and Tables

**Figure 1 polymers-14-01715-f001:**
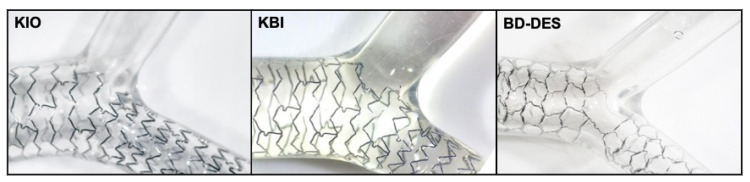
Optical images of platforms implanted into a model of bifurcation.

**Figure 2 polymers-14-01715-f002:**
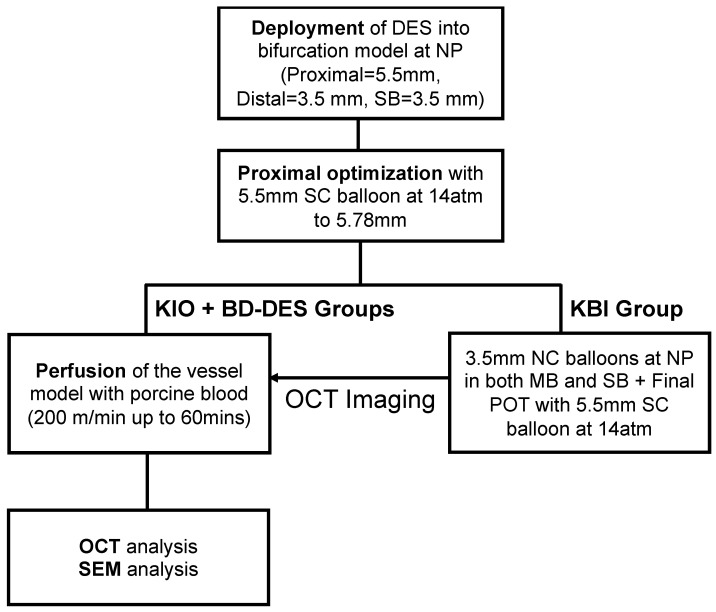
Flowchart of benchtop study using in vitro bifurcation model for the three groups.

**Figure 3 polymers-14-01715-f003:**
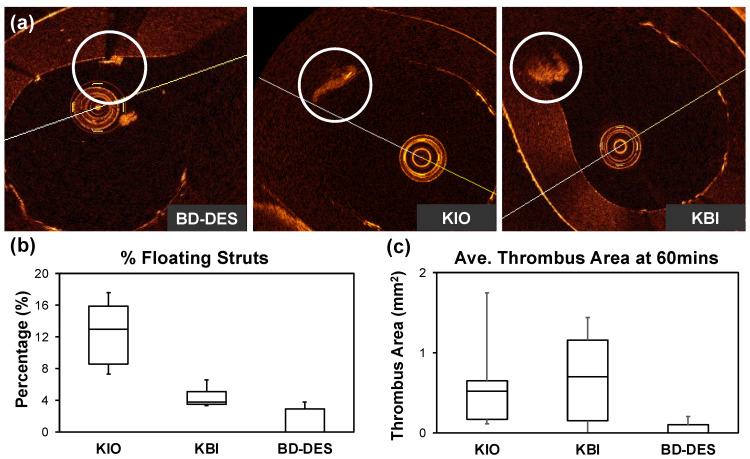
(**a**) Representative OCT images of thrombus formation on the stents at 60 mins; (**b**) percentage of floating struts and (**c**) average thrombus area based on OCT quantification, *n* = 5.

**Figure 4 polymers-14-01715-f004:**
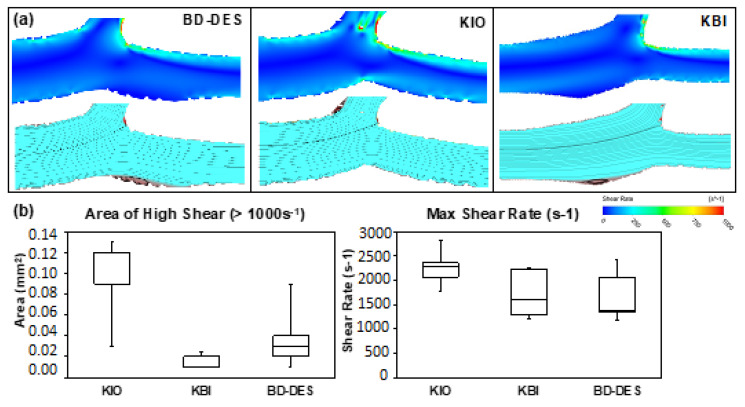
(**a**) Representative CFD images and (**b**) area of high shear (>1000 s^−1^) and maximum shear rate quantification of the three groups (*n* = 5).

**Figure 5 polymers-14-01715-f005:**
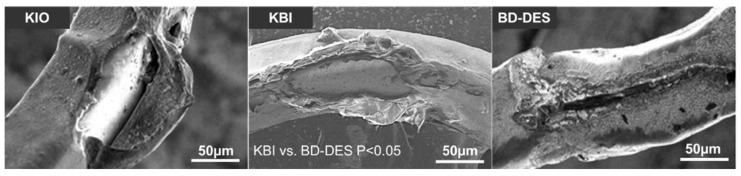
Representative SEM images of drug coating damage. The image was taken at ×500 magnification and scale bar = 50 μm.

**Table 1 polymers-14-01715-t001:** OCT analysis of stents overexpansion (*n* = 5).

	KIO	KBI	BD-DES	*p* Value
**OCT Proximal Diameter (mm)**				
Min	5.40 (5.23–5.58)	5.36 (5.12–5.52)	5.33 (5.23–5.37)	>0.05
Mean	5.55 (5.42–5.69)	5.52 (5.30–5.66)	5.50 (5.40–5.53)	>0.05
Max	5.75 (5.60–5.79)	5.65 (5.41–5.75)	5.59 (5.55–5.64)	>0.05
**OCT Proximal Lumen Area (mm^2^)**				
Area	24.22 (23.09–25.4)	23.93 (22.06–25.16)	23.81 (22.92–24.00)	>0.05
**OCT Proximal Eccentricity Index**				
EI	1.06 (1.05–1.06)	1.05 (1.04–1.06)	1.05 (1.04–1.05)	>0.05
**OCT Strut Analysis**				
WA (%)	78.6 (76.1–85.3)	93.3 (88.1–93.4)	94.3 (88.5–98.8)	>0.05
Floating (%)	15.9 (13.0–17.7)	3.8 (3.5–5.1)	0.0 (0.0–2.7)	>0.05
MA (%)	4.9 (4.7–5.3)	3.3 (1.6–6.0)	2.7 (1.2–2.9)	>0.05

**Table 2 polymers-14-01715-t002:** SEM analysis of polymer coating damage (*n* = 5).

SEM Coating Analysis				
	KIO	KBI	BD-DES	*p* Value
Category 1	19 (11–21)	17 (15–27)	7 (6–14)	>0.05
Category 2	7 (7–11)	21 (18–21)	1 (0–3)	<0.05 ^b^
Category 3	4 (3–12)	11 (7–12)	0 (0–1)	<0.05 ^a,b^
Category 4	5 (3–15)	20 (16–21)	0 (0–0)	<0.05 ^b^

Values are presented as median (interquartile range). ^a^ if *p* < 0.05 from KIO, ^b^ If *p* < 0.05 from KBI.

## Data Availability

The data presented in this study are available on request from the corresponding author.

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
