# Peer review of "Polymer Coating Integrity, Thrombogenicity and Computational Fluid Dynamics Analysis of Provisional Stenting Technique in the Left Main Bifurcation Setting: Insights from an In-Vitro Model"

_polymers, 2022, doi:10.3390/polym14091715_

Round 1

Reviewer 1 Report

Please consider the the following points to improve the quality of the manuscript:

  1.  ''After a specified period of time, the polymer slowly degrades into carbon dioxides and water.'' please mention the reference.
  2. ''Benchtop flow loop systems set by various groups showed the advantages and reproducibility of this method to rate different stents and stenting methods [19,20,21,22,23].'' please explain the advantages following the references.
  3. Is it possible to characterize the polymer damage by spectroscopic analysis (FTIR analysis)

Reviewer 2 Report

The latest generation of drug-eluting stents (DES) were compared by computational fluid dynamics (CFD), OCT imaging and scanning electron microscopy (SEM) analysis. Here are a few suggestions to improve your writing and understanding.

  1. In section 3.2 Computational Fluid Dynamics only includes CFD images and areas of high shear (>1000s-1) and maximum shear rate quantification of the three groups. The analysis of CFD images and the difference analysis and discussion of shear forces between the three groups are missing.
  2. this article lacks the analysis part and the supplement of material and equipment information.
  3. There is a lack of description of the verification of the correctness of the numerical simulations in terms of the results of the tests.

Round 2

Reviewer 1 Report

Please consider the following points to improve the manuscript

  1. Briefly describe the polymers (acrylic fluoropolymer) biodegradability for stenting in introduction.
  2.  Although the results are promising, there was coating damage (line 346); please add your opinion to overcome this limitation.

Author Response

This manuscript is a resubmission of an earlier submission. The following is a list of the peer review reports and author responses from that submission.

Round 1

Reviewer 1 Report

The latest generation of drug-eluting stents (DES) were compared by computational fluid dynamics (CFD), OCT imaging and scanning electron microscopy (SEM) analysis, and the implantation of KIO without collateral expansion (SB) versus KBI and bifurcation specific drug-eluting stents (BD-DES). Here are a few suggestions to improve your writing and understanding.

  1. In section 2.5 of the article.The Computational Fluid Dynamics, which builds a 2D model of the scaffold, indicates the setting of the conditional parameters, but does not explain the correlation between the grid and the Computational results.
  2. In section 3.1 of OPTICAL coherence tomography, numerical differences between KBI, DB-DES and KIO were explained in terms of the percentage of WA struts, the percentage of floating struts, the percentage of MA struts, and the thrombotic area, but the analysis of the differences in these numerical results was missing in the discussion.
  3. In section 3.2 Computational Fluid Dynamics only includes CFD images and areas of high shear (>1000s-1) and maximum shear rate quantification of the three groups. The analysis of CFD images and the difference analysis and discussion of shear forces between the three groups are missing.
  4. As for the discussion and analysis part of the paper, it is believed that the thrombosis area of BD-DES is smaller, which is related to the stent structure. The composition of the two connecting rods reduces the proportion of metal to artery, making the lumen less damaged. This result is open to debate and argument.
  5. As for the reference part of the paper, the latest research in the past five years is added as the theoretical basis of the paper, which makes the paper more scientific.

In a word, this article lacks the analysis part and the supplement of material and equipment information. 

Reviewer 2 Report

The manuscript is interesting. Please consider the following points to improve the manuscript;

  1. Please add a separate paragraph about polymer coating suitability and challenges for new-generation DES implantation with KBI and KIO approach, particularly previous published work.
  2. In Figure 5, please mention the bar area in SEM photograph, its not clear.
  3. If possible, please mention the polymer name clearly in experimental section
  4. Please mention the coating preparation? is it commercially available, how you applied the coating?